# Low Muscle Mass as a Prognostic Factor for Early Postoperative Outcomes in Pediatric Patients Undergoing the Fontan Operation: A Retrospective Cohort Study

**DOI:** 10.3390/jcm8081257

**Published:** 2019-08-19

**Authors:** Jimi Oh, Won-Jung Shin, DaUn Jeong, Tae-Jin Yun, Chun Soo Park, Eun Seok Choi, Jae Moon Choi, Mijeung Gwak, In-Kyung Song

**Affiliations:** 1Department of Anesthesiology and Pain Medicine, Asan Medical Center, University of Ulsan College of Medicine, 88 Olympic-ro 43-gil, Songpa-gu, Seoul 05505, Korea; 2Department of Anesthesiology and Pain Medicine, Laboratory for Cardiovascular Dynamics, Asan Medical Center, University of Ulsan College of Medicine, 88 Olympic-ro 43-gil, Songpa-gu, Seoul 05505, Korea; 3Department of Pediatric Cardiac Surgery, Asan Medical Center, University of Ulsan College of Medicine, 88 Olympic-ro 43-gil, Songpa-gu, Seoul 05505, Korea

**Keywords:** child, heart defects, congenital, fontan procedure, postoperative complications, sarcopenia

## Abstract

The impact of low muscle mass on pediatric cardiac patients remains unclear. We investigated the impact of low muscle mass on early postoperative outcomes in patients undergoing the Fontan operation. The electronic medical records of 74 patients (aged <18 years) who underwent the Fontan operation were retrospectively reviewed. The cross-sectional areas of the erector spinae and pectoralis muscles were measured using preoperative chest computed tomography (CT), normalized to the body surface area, and combined to obtain the total skeletal muscle index (TSMI). Low muscle mass was defined as a TSMI value lower than the median TSMI for the second quintile. The incidence of major postoperative complications was higher in patients with low muscle mass than in those with high muscle mass (48% (15/31) versus 14% (6/43); *P* = 0.003). Multivariable analyses revealed that a higher TSMI was associated with a lower likelihood of an increased duration of intensive care unit (>5 days) and hospital stay (>14 days) (odds ratio (OR) 0.86; 95% confidence interval (CI) 0.77–0.96; *P* = 0.006 and OR 0.92; 95% CI 0.85–0.99; *P* = 0.035 per 1 cm^2^/m^2^ increase in TSMI) and incidence of major postoperative complications (OR 0.90; 95% CI 0.82–0.99; *P* = 0.039 per 1 cm^2^/m^2^ increase in TSMI). Preoperative low muscle mass was associated with poor early postoperative outcomes in pediatric patients undergoing the Fontan operation.

## 1. Introduction

Owing to advancements in surgical techniques, interventional procedures, and perioperative management, the survival of pediatric patients with functional single ventricle physiology has dramatically improved through the use of a series of staged palliative procedures, which are initiated during the neonatal period. This improved survival has led to an increased number of patients undergoing the Fontan operation [1]. However, it should be noted that the Fontan operation only partially substitutes the normal circulation and does not completely correct the anatomical variations of the functional single ventricle. Therefore, patients who undergo the Fontan operation and exhibit long-term survival might experience clinical changes such as a loss of skeletal muscle mass.

Sarcopenia is characterized by a decrease in the quality and quantity of skeletal muscle mass, which can lead to worsened quality of life and mortality. Low muscle mass has a considerable impact on disease severity and mortality in adults with cardiopulmonary diseases [2]. Moreover, malnutrition and sarcopenia are associated with worse outcomes in pediatric patients with severe chronic illness [3]. These data suggest that low muscle mass can be associated with postoperative complications and mortality in pediatric patients who undergo the Fontan operation. If low muscle mass considerably affects the postoperative outcomes in pediatric patients undergoing the Fontan operation, it could potentially serve as a prognostic factor to complement Choussat’s “Ten Commandments”, which are used to identify a potential Fontan candidate [4].

Thus, this study aimed to evaluate the skeletal muscle mass using chest computed tomography (CT) in pediatric patients undergoing the Fontan operation. In addition, we investigated whether preoperative low muscle mass could be used to predict early postoperative outcomes in pediatric patients undergoing the Fontan operation.

## 2. Materials and Methods 

### 2.1. Study Participants and Measurements

This retrospective cohort study was approved by the Institutional Review Board and Institutional Ethics Committee of the Asan Medical Center, Seoul, South Korea on 18 August 2018 (No 2018-0982). The need for individual informed consent was waived by the Institutional Review Board. The electronic medical records of patients aged <18 years who underwent the Fontan operation between January 2006 and December 2016 at the Asan Medical Center were reviewed. Exclusion criteria were patients with congenital anomalies other than functional single ventricle, growth disturbances and eating disorders affecting normal growth, patients who underwent other operations in addition to cardiac surgery, and patients without preoperative chest CT scans.

### 2.2. Clinical Variables and Postoperative Outcomes

Baseline demographic variables included age, sex, weight, height, body surface area (BSA), birth weight, and premature birth (gestational age, <37 weeks). Intraoperative variables included anesthesia duration, surgical duration, cardiopulmonary bypass (CPB) duration, and maximum vasoactive–inotropic score (VIS_max_) after weaning from CPB. VIS was calculated using the method described by Gaies et al. [5]. The postoperative outcome variables of interest were the duration of mechanical ventilation, the duration of intensive care unit (ICU) and hospital stays, postoperative VIS_max_ during the first 48 h, postoperative complications, and 30-day mortality. Postoperative complications that were examined included the following: reoperation or readmission within 30 days, cardiovascular collapse, cardiopulmonary resuscitation, extracorporeal membrane oxygenation or permanent pacemaker implantation, seizures, focal neurological deficit, pneumonia, and acute respiratory failure. All the postoperative complications were recorded and individually scored according to the Clavien–Dindo classification, which comprised five grades: grade 1 (any deviation from the normal postoperative course without the need for pharmacological treatment or surgical and radiological interventions); grade 2 (requiring pharmacological treatment); grade 3 (requiring surgical or radiological intervention); grade 4 (life-threatening complications requiring intensive care/ICU management); and grade 5 (patient death) [6]. A postoperative complication of grade 3 or higher was defined as a major postoperative complication.

### 2.3. Skeletal Muscle Mass Measurement Using Chest CT and the Definition of Low Muscle Mass

Our institution conducts a computed tomography (CT) scan as a routine assessment before the Fontan operation. However, if necessary, cardiac catheterization is also performed to obtain information. All the preoperative chest CTs were obtained within the three months prior to the operation. Cardiothoracic CT was performed using a 128-slice dual-source scanner (SOMATOM Definition Flash; Siemens Healthineers, Forchheim, Germany). For scanning, the z-flying focal spot technique was employed using 2 × 64 × 0.6-mm slices with a gantry rotation time of 280 ms, temporal resolution of 75 ms, slice width of 0.75 mm, and reconstruction interval of 0.4 mm during free breathing in all the pediatric patients [7]. Skeletal muscle was quantified using a Hounsfield unit range of −29 to +150. For the skeletal muscle measurements using CT images, we used the method described by Miller et al., in which the skeletal muscle mass is estimated by manually tracing the borders of the right and left pectoralis muscles (PMs) and erector spinae muscles (ESMs) within 1 cm of the sternoclavicular joint and at the 12th thoracic vertebral level (T12), respectively (Figure 1) [8].

The cross-sectional areas of the PM and ESM were normalized to the BSA, which resulted in the PM index (PMI) and ESM index (ESMI) [2]. BSA was calculated using the Haycock formula:BSA (m^2^) = weight (kg)^0.5378^ × height (cm)^0.3964^ × 0.024265(1)

The sum of PMI and ESMI provided the total skeletal muscle index (TSMI). All the CT images were analyzed by two observers who were trained by a radiologist specializing in musculoskeletal imaging. Both the observers were blinded to the clinical data. Intraobserver and interobserver variability were estimated using the intraclass correlation coefficient. 

Given that there is no cutoff value or diagnostic criteria for defining low muscle mass in pediatric patients, the patients were divided into quintiles according to the TSMI values. To investigate the effect of TSMI on early postoperative outcomes, the patients were categorized into two groups: low muscle mass and high muscle mass. Low muscle mass was defined as those with a TSMI value lower than the median TSMI for the second quintile.

### 2.4. Statistical Analysis

Data are presented as mean ± SD, median (interquartile range, IQR), and number (%), as appropriate. In case of missing data, either the variable or the case was excluded from the analysis. Categorical variables were analyzed using Pearson’s chi-squared (*χ*^2^) test or Fisher’s exact test. Continuous variables were analyzed using Student’s *t*-test or the Mann–Whitney *U* test. Poor postoperative outcomes were defined as any incidence of major postoperative complications, duration of ICU stay >5 days, or duration of hospital stay >14 days. A univariate logistic regression analysis was used to determine the association between low muscle mass and poor postoperative outcomes. A multivariable logistic regression analysis was performed after adjusting for the clinical variables that showed significant differences. For all analyses, a *P* value of <0.05 was considered statistically significant. All the analyses were conducted using the Statistical Package for the Social Sciences software (IBM^®^ SPSS^®^ Statistics 23, SPSS Inc., IBM Corporation, Armonk, NY, USA).

## 3. Results

### 3.1. Study Population

Of the 194 pediatric patients who underwent the Fontan operation between January 2006 and December 2016, 74 were included in the final analysis (Figure 2). The baseline demographic and clinical characteristics of the study patients are shown in Table 1.

### 3.2. Low Muscle Mass Assessment

The intraclass correlation coefficient for CT measurements was 0.986 (*P* < 0.001), which was in the acceptable range.

TSMI histograms of the study population are shown in Figure 3. Overall, 31 (42%) patients met the criteria for low muscle mass, which was indicated when the TSMI decreased to <19.3 cm^2^/m^2^.

### 3.3. Correlation Between Low Muscle Mass and Early Postoperative Complications

The perioperative characteristics of patients with and without low muscle mass are shown in Table 1. No significant differences were observed between the two groups regarding age, sex, BSA, birth weight, and premature birth. Among the 74 patients evaluated, 31 (42%) patients developed postoperative complications, and of these complications, 21 (28%) were classified as major postoperative complications (Clavien–Dindo grade ≥3). The incidence of major postoperative complications was significantly higher in patients with low muscle mass than in those with high muscle mass (48% (15/31) versus 14% (6/43); *P* = 0.003). Patients with low muscle mass were more likely to be associated with a longer duration of ICU stay compared with those with high muscle mass (4 (2–9) days versus 3 (2–4) days; *P* = 0.009) (Table 1). No patient died within 30 days of the Fontan operation.

### 3.4. Association between Low Muscle Mass and Poor Postoperative Outcomes

The results of the univariate and multivariable regression analyses are shown in Table 2. The multivariable linear regression model, after adjusting for age, sex, weight, premature birth, and CPB duration revealed that the TSMI was independently associated with an increased duration of ICU and hospital stay (odds ratio (OR) 0.86; 95% confidence interval (CI) 0.77–0.96; *P* = 0.006 and OR, 0.92; 95% CI, 0.85–0.99; *P* = 0.035, per 1 cm^2^/m^2^ increase in TSMI) and incidence of major postoperative complications (OR 0.90; 95% CI 0.82–0.99; *P* = 0.039, per 1 cm^2^/m^2^ increase in TSMI).

## 4. Discussion

This study demonstrated that preoperative low muscle mass as defined by TSMI was associated with an increased duration of ICU and hospital stay and incidence of major postoperative complications in pediatric patients undergoing the Fontan operation. To the best of our knowledge, this is the first study to show that the TSMI measured from chest CT images might be a useful tool for predicting early postoperative outcomes in pediatric patients undergoing the Fontan operation. 

Previous studies identifying the predictors of outcomes after the Fontan operation have traditionally focused on single ventricle-specific anatomical factors [9,10]. These studies demonstrated that ventricular morphology, the type of procedure performed, and patient age at the time of the procedure were associated with survival rates [11,12]. However, given the Fontan operation is associated with improved survival rates in pediatric patients, the focus of research has shifted toward the long-term consequences of the Fontan physiology. In addition, recent research has reinforced the suggestion that Fontan physiology is not simply a cardiovascular disease but rather a multi-organ disease with complex interactions between the cardiovascular and non-cardiovascular organs [1]. These results help improve our understanding of non-cardiac sequelae and the perioperative clinical course of the Fontan physiology during the development and growth of pediatric patients. A multidisciplinary approach is clearly important to identifying and managing the corrective factors involved in the postoperative outcomes of pediatric patients who undergo the procedure.

Sarcopenia, which can be literally defined as “poverty of flesh”, may occur at an early age in patients with chronic illness. Sarcopenia represents the physical component of a multifactorial syndrome that results in a progressive loss of skeletal muscle mass. Several reports have shown that the loss of skeletal muscle mass in critically ill pediatric patients with acute lymphoblastic leukemia [13] and pediatric patients who undergo liver transplantation [3] is associated with increased rates of hospitalization. Furthermore, children with congenital heart diseases (CHDs) are at a risk of poor growth and failure to thrive [3]. Particularly, pediatric patients with single ventricle physiology who undergo cardiac surgery experience significant nutritional and metabolic stresses [14]. Although children with CHDs typically display normal birth weight for gestational age, younger children with CHDs often present with impaired growth parameters, which are associated with an increased duration of hospital stay [15]. Therefore, it is crucial to accurately determine and optimize their nutritional status before and after palliative and corrective surgical procedures. Nevertheless, few studies have evaluated the preoperative nutritional status in children with CHDs, except for studies that reported commonly used anthropometric parameters such as weight-for-height, height-for-age, or body mass index [15]. In addition, a previous meta-analysis [16] and a European multicenter study [17] did not recommend using a single screening method for determining nutritional status, because one method is not sufficient to determine a patient’s nutritional status. In the present study, we considered that measuring the skeletal muscle mass using CT images would be a useful method for objectively assessing the preoperative nutritional status in pediatric patients undergoing the Fontan operation.

Our results demonstrated that the incidence of major postoperative complications (Clavien–Dindo grade ≥3) was higher in patients with low muscle mass than in those without low muscle mass. Also, regarding multivariable analyses, preoperative TSMI was independently associated with an increased incidence of major postoperative complications. These data suggest that the degree of low muscle mass, which can be easily determined using cross-sectional CT images, serves as an objective measure to identify patients with a risk of major postoperative complications following the Fontan operation.

The mechanisms by which low muscle mass leads to an increased risk of major postoperative complications remain unclear. However, several mechanisms can be hypothesized. First, skeletal muscle mass is an important factor in determining cardiac performance, given that it helps drive blood toward the heart and increases stroke volume. Therefore, patients who undergo the Fontan operation and who are preload-sensitive might be affected by low muscle mass during the perioperative period. Second, in adults with the Fontan physiology, reduced skeletal muscle mass decreases the aerobic capacity of skeletal muscles and phosphocreatine levels, which are important indicators of myocellular energy status [18]. Additionally, increased catabolic stress due to reduced skeletal muscle mass results in glucose intolerance, which increases inflammatory cytokine levels and chronotropic incompetence. These factors might further contribute to the underlying pathophysiology of unfavorable postoperative complications [19]. These results emphasize that reduced skeletal muscle mass is a prognostic factor for predicting postoperative outcomes in pediatric patients with a functional single ventricle, and might play a role in overall health status as well as the ability to tolerate acute stress due to the Fontan operation. Therefore, the diagnostic criteria for an ideal Fontan candidate could include information regarding low muscle mass in addition to the degree of atrioventricular and aortopulmonary collateral flow, as well as abnormalities in ventricular rhythm and/or function.

Currently, there is no standard method or numerical value that defines low muscle mass in pediatric patients. A recent European Consensus Statement has recognized CT as the gold standard for the non-invasive assessment of muscle mass in adult patients [20]. Most studies using CT images for evaluating skeletal muscle mass have measured the cross-sectional area of muscle at the lumbar spine level using abdominal CT images [21]. However, for pediatric patients undergoing cardiac procedures, including the Fontan operation, it is much more practical to use existing thoracic CT images rather than to obtain additional abdominal CT images. Nemec et al. have demonstrated a correlation between skeletal muscle indices at the L3 and T12 levels and have suggested that muscle mass measured at the T12 level should be used to diagnose low muscle mass using chest CT alone [21]. Therefore, we chose to examine PM and ESM due to the availability of chest CT images obtained during the preoperative evaluation. 

Our study has several limitations. First, it was a retrospective analysis of patients’ medical records that measured muscle mass, but not muscle function, to confirm sarcopenia. In future studies, both muscle mass and function should be evaluated to determine the impact of sarcopenia on postoperative outcomes in pediatric patients with CHDs. Second, the cutoff value used for defining low muscle mass using chest CT was not derived from healthy children. However, we determined that dividing patients into quintiles and classifying low muscle mass is a clinically meaningful method, as in the previous study [22]. In addition, given that CT is rarely indicated for healthy children, validated cutoff values for the normal pediatric population based on age, sex, and muscle type are still lacking. Third, hemodynamic or echocardiographic parameters were not considered in the analysis, because the data were not available at the time when the chest CT scans were taken. Given that hemodynamic burden might affect both low muscle mass and postoperative outcomes and that there is little research on the association between hemodynamic parameters and low muscle mass in pediatric patients with CHDs, further studies are needed to identify these parameters. Finally, we performed a post-hoc power analysis for the sample size, given the retrospective design of this study. The incidence of major postoperative complications in all of the study patients was 28%, and that in the low muscle mass group was 48%. The sample size of 74 resulted in a post-hoc power of 95.4%, considering an α-error of 0.05.

## 5. Conclusions

Preoperative low muscle mass was associated with an increased incidence of major postoperative complications in pediatric patients undergoing the Fontan operation. Therefore, preoperative TSMI obtained from chest CT imaging might be a useful tool for predicting early postoperative outcomes after the operation.

## Figures and Tables

**Figure 1 jcm-08-01257-f001:**
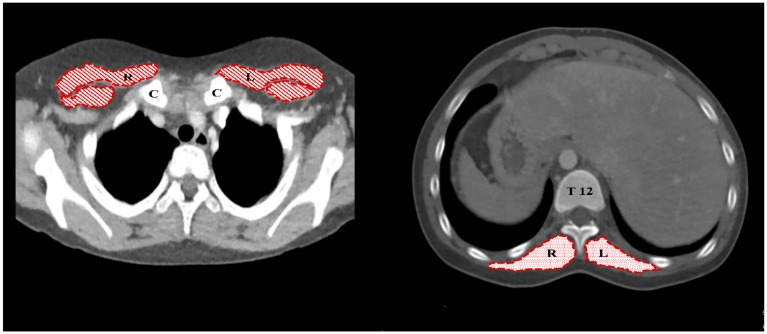
Axial computed tomography (CT) image for the cross-sectional area measurements of pectoralis and erector spinae muscles. The areas shaded with oblique lines are the pectoralis muscles (left), and the areas shaded with dots are the erector spinae muscles (right). R, right; L, left; C, clavicle; T12, 12th thoracic vertebra.

**Figure 2 jcm-08-01257-f002:**
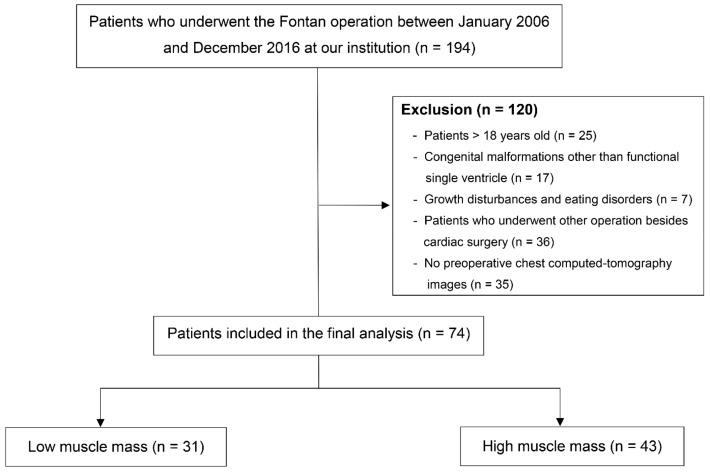
Flowchart of the patient selection and classification.

**Figure 3 jcm-08-01257-f003:**
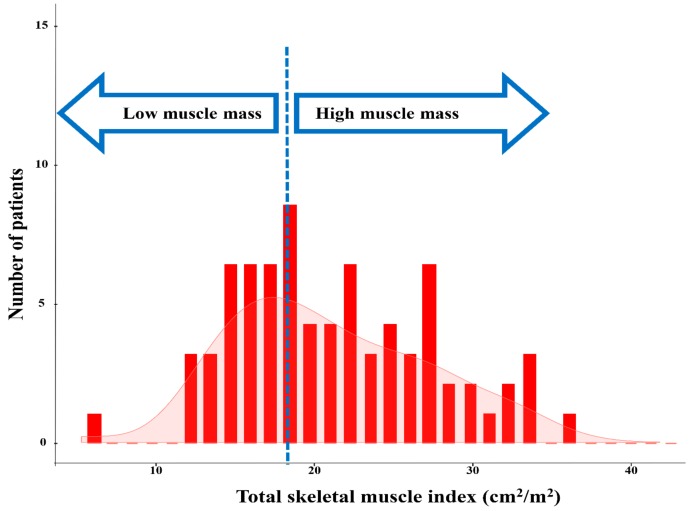
Distribution of the total skeletal muscle index in the pediatric patients undergoing the Fontan operation (red bars).

**Table 1 jcm-08-01257-t001:** Clinical characteristics of 74 pediatric patients who underwent the Fontan operation.

	Total (*n* = 74)	Low Muscle Mass (*n* = 31)	High Muscle Mass (*n* = 43)
Demographic Variables			
Age, year	2.9 (2.8–3.3)	3.0 (2.8–3.0)	3.0 (2.7–3.5)
Sex, male	38 (51)	16 (52)	22 (51)
Weight, kg	14.0 (13.0–15.6)	13.9 (12.8–14.8)	14.2 (13.1–16.2)
Height, cm	93.3 (91.3–96.1)	93.3 (91.7–94.8)	93.4 (91.2–96.9)
Body surface area, m^2^	0.60 (0.58–0.64)	0.60 (0.58–0.65)	0.60 (0.57–0.68)
Birth weight, kg	3.17 (2.80–3.46)	3.10 (2.73–3.37)	3.19 (2.84–3.49)
Premature birth	11 (15)	4 (13)	7 (16)
Computed Tomography Measurements			
PMI, cm^2^/m^2^	13.1 (9.8–15.9)	9.5 (9.0–10.8)	15.8 (13.1–17.2) *
ESMI, cm^2^/m^2^	8.6 (6.5–10.5)	6.3 (5.6–7.0)	10.4 (8.9–11.7) *
TSMI, cm^2^/m^2^	21.7 (16.9–26.4)	16.5(14.7–17.8)	24.9 (22.2–28.8) *
Intraoperative Variables			
Anesthesia duration, min	391 ± 112	410 ± 149	379 ± 77
Surgical duration, min	331 ± 107	348 ± 140	318 ± 75
CPB duration, min	110 ± 78	128 ± 117	99 ± 34
VIS_max_^1^	5.0 (5.0–9.4)	7.0 (5.0–10.0)	5.0 (5.0–8.0) *
Postoperative Variables			
VIS_max_^2^	10.0 (5.5–12.5)	10.0 (7.0–17.5)	8.0 (5.0–10.0)
Duration of MV, h	20 (13–43)	22 (16–53)	17 (13–33)
ICU stay, days	3 (2–7)	4 (2–9)	3 (2–4) *
Hospital stay, days	24 (12–30)	15 (12–45)	15 (12–20)
Major postoperative complications	21 (28)	15 (48)	6 (14) *

Values are mean ± SD, median (interquartile range, IQR), or number (proportion). PMI, pectoralis muscle index; ESMI, erector spinae muscle index; TSMI, total skeletal muscle index; CPB, cardiopulmonary bypass; VIS_max_^1^, maximum vasoactive-inotropic score after weaning from CPB; VIS_max_^2^, maximum vasoactive-inotropic score during postoperative 48 h; MV, mechanical ventilation; ICU, intensive care unit. Premature birth was defined as gestational age <37 weeks. Major postoperative complications were defined as Clavien–Dindo classification grade 3 or higher. * *P* <0.05 vs. low muscle mass group.

**Table 2 jcm-08-01257-t002:** Univariate and multivariable analyses of risk factors for poor postoperative outcomes after the Fontan operation.

	Univariate		Multivariable	
	OR (95% CI)	*P*	OR (95% CI)	*P*
Major Postoperative Complications				
Age, year	0.70 (0.40–1.21)	0.202		
Sex, male	0.62 (0.20–2.00)	0.428		
Body weight, kg	1.24 (0.97–1.60)	0.087		
Premature birth	1.93 (0.35–10.6)	0.447		
CPB duration, min	1.00 (0.99–1.01)	0.432		
TSMI, cm^2^/m^2^	0.89 (0.81–0.99)	0.031	0.90 (0.82–0.99)	0.039
Duration of ICU Stay > 5 Days				
Age, year	1.17 (0.66–2.06)	0.585		
Sex, male	1.56 (0.47–5.13)	0.156		
Body weight, kg	0.88 (0.69–1.12)	0.306		
Premature birth	4.97 (0.52–47.1)	0.162		
CPB duration, min	1.00 (1.00–1.01)	0.318		
TSMI, cm^2^/m^2^	0.87 (0.78–0.97)	0.014	0.86 (0.77–0.96)	0.006
Duration of Hospital Stay > 14 days				
Age, year	1.35 (0.88–2.08)	0.175		
Sex, male	0.90 (0.31–2.61)	0.841		
Body weight, kg	0.88 (0.73–1.05)	0.882		
Premature birth	2.13 (0.52–8.75)	0.292		
CPB duration, min	1.00 (1.00–1.01)	0.444		
TSMI, cm^2^/m^2^	0.92 (0.85–0.99)	0.039	0.92 (0.85–0.99)	0.035

OR, odds ratio; CI, confidence interval; CPB, cardiopulmonary bypass; TSMI, total skeletal muscle index. Premature birth was defined as gestational age <37 weeks. Major postoperative complications were defined as Clavien–Dindo classification grade 3 or higher.

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
