# Peer review of "Low Muscle Mass as a Prognostic Factor for Early Postoperative Outcomes in Pediatric Patients Undergoing the Fontan Operation: A Retrospective Cohort Study"

_jcm, 2019, doi:10.3390/jcm8081257_

Round 1
Reviewer 1 Report
This article described sarcopenia as a prognostic factor for the Fontan operation comparing it to the Nuss operation (healthy controls). The authors found that sarcopenia was more common in Fontan patients, which is not surprising and that sarcopenia was an independent predictor of poor early post op outcomes in Fontan patients.
I have a few questions:
How were CT sans performed? As part of routine clinical practice? This is a retrospective study so I assume they were not performed just to look at muscle mass (for which a specific IRB would be needed and would also raise ethical questions).
If CT scans are performed as routine practice, why is that? Is that in leu of cardiac catheterizations?
Can the same information be obtained without radiation, e.g. via MRI?
The authors control for a number of factors, however there is no mention of hemodynamic or echocardiographic parameters. This is very important. Can the authors control for factors such as ventricular function, AV valve regurgitation, any residual lesion, elevated Glenn pressures, elevated EDP, PA size, arch obstruction, rhythm disturbances etc? Just to make sure the hemodynamic burden is not why the patients with sarcopenia have poor outcomes? It would also address another question, as to whether the hemodynamic state is what results in sarcopenia.
Author Response
Response to Reviewer 1 Comments
This article described sarcopenia as a prognostic factor for the Fontan operation comparing it to the Nuss operation (healthy controls). The authors found that sarcopenia was more common in Fontan patients, which is not surprising and that sarcopenia was an independent predictor of poor early post op outcomes in Fontan patients.
I have a few questions:
Point 1. How were CT scans performed? As part of routine clinical practice? This is a retrospective study so I assume they were not performed just to look at muscle mass (for which a specific IRB would be needed and would also raise ethical questions).
Response 1: Thank you for the valuable and positive comments. According to the policy of the department of pediatric cardiology and pediatric cardiac surgery, all patients undergo chest CT examination as a part of routine preoperative evaluation before the Fontan operation in our institution. Therefore, no specific IRB regarding Chest CT was needed. We added this content in the methods section as follows.
“At our institution, pediatric patients scheduled for the Fontan operation undergo chest CT scan as a part of routine preoperative evaluation.”
Point 2. If CT scans are performed as routine practice, why is that? Is that in leu of cardiac catheterizations?
Response 2: Although cardiac catheterization is one of the most important cardiac imaging modalities, it has limitations due to its invasive nature. The limitations of cardiac catheterization include the overlapping of great vessels, difficulty in demonstrating systemic and pulmonary vessels at the same time, catheter-related complications, and high doses of iodinated contrast material and ionizing radiation. Common complications of cardiac catheterization include thrombosis, arteriovenous fistula, pseudoaneurysm formation, hemorrhage, and dissection. Moreover, the repeated puncture of femoral vessels correlates with femoral vessel occlusion and asymmetric growth of bilateral legs and postprocedural monitoring may take several hours in a recovery ward. Cardiac catheterization may require general anesthesia [J Formos Med Assoc. 2015 Nov;114(11):1061-8]. Given that CT can overcome these limitations, recent studies have reported a dramatic increase in CT utilization among children with congenital heart diseases. Furthermore, pediatric cardiac CT does not require the same level of deep sedation as cardiac catheterization, requiring only adequate peripheral venous access. Therefore, our institution conducts a CT scan as a routine assessment before the Fontan operation. However, if necessary, cardiac catheterization is also performed to obtain information.
Point 3. Can the same information be obtained without radiation, e.g. via MRI?
Response 3: As the reviewer pointed out, CT and MRI are considered standard methods for noninvasive evaluation of muscle mass [BMC Geriatr. 2016 Oct 5;16(1):170]. However, MRI has some disadvantages in pediatric patients with CHD compared with CT. MRI takes more time than CT, so it is difficult to perform in hemodynamically unstable pediatric patients and the need for general anesthesia increases. Also, low spatial resolution and high cost are considered as limitation of MRI in the measurement of skeletal muscle mass on a routine procedure.
Point 4. The authors control for a number of factors, however there is no mention of hemodynamic or echocardiographic parameters. This is very important. Can the authors control for factors such as ventricular function, AV valve regurgitation, any residual lesion, elevated Glenn pressures, elevated EDP, PA size, arch obstruction, rhythm disturbances etc? Just to make sure the hemodynamic burden is not why the patients with sarcopenia have poor outcomes? It would also address another question, as to whether the hemodynamic state is what results in sarcopenia.
Response 4: Thank you for this thoughtful comment. We agree with the reviewer that hemodynamic burden may affect patients’ outcomes. We understand the potential limitation of the current study on the issue raised by the reviewer and added this shortcoming in the limitation section more clearly. Now it reads in the limitation section as follows.
“Third, hemodynamic or echocardiographic parameters were not considered in the analysis because the data were not available at the time when the chest CT scans were taken. Given that hemodynamic burden might affect both muscle wasting and postoperative outcomes and that there is little research on the association between hemodynamic parameters and muscle wasting in pediatric patients with CHDs, further studies are needed to identify these parameters.”

Reviewer 2 Report
This manuscript presents results from a retrospective study providing evidence that the use of chest CT images to determine muscle mass may be beneficial for predicting early postoperative outcomes in pediatric patients undergoing Fontan procedures. The results have potential clinical significance and the article is likely to be of interest to your readership. I have the following major comments regarding the manuscript:
1. Sarcopenia by definition is the age-related loss of skeletal muscle mass and function. It is a geriatric condition and therefore not observed in children. Furthermore, the investigators have only examined muscle mass, and not function. Therefore the title, abstract and wording throughout the paper should refer only to low muscle mass and not sarcopenia.
2. I do not see the value of including a control group of children who underwent the Nuss procedure. These patients do not seem to be comparable to the Fontan group. The Nuss group is a mean of 5 years older than Fontan (mean age 8 vs 3 years), and the upper age range for Fontan falls below the lower age range for Nuss. Although muscle mass estimates are normalised to body surface area, there is still an assumption that proportions of muscle mass are similar in younger children and older children, which may not be the case. Age-adjusted results are provided but this potentially doesn't control for all the factors that may explain better muscle mass in older children.
3. Further to the above two points, I do not believe it is appropriate to define low muscle mass cut-offs using the Nuss group as the reference population. In addition to the fact that they do not appear to be comparable populations, existing low muscle mass cut-offs for older adults are generally based on values around ~<2SD below the mean for healthy young adult populations. The authors of this manuscript have used the median for older Nuss patients to identify low muscle mass in younger Fontan patients, and this is not based on evidence. I therefore believe the analyses should focus on continuous values for muscle mass rather than arbitrary dichotomous variables
4. In summary, I do not believe that the data for the Nuss group adds value to the paper. It therefore could be removed so that the focus is on the Fontan group only. If cut-offs were to be used for low muscle mass, I think it would be of more interest to determine cut-offs associated with poorer prognosis from within the Fontan population; this data would be of greater utility to clinicians.
Other comments:
5. The odds ratios in the abstract and table need to include units for the independent variables.
6. The abstract should state that this is a retrospective study.
7. Is data available on the intra- and inter-operator reliability of the CT measurements?
8. Was a power calculation performed to determine whether the sample size is adequate?
9. There appears to be an error in the description of the study population beginning on line 125; it states that "194 pediatric patients who underwent the Fontan operation between January 2006 and December 2016, 135 were included in the study". But the 194 appears to be for both Fontan and Nuss, as only 74 Fontan were included.
Author Response
Response to Reviewer 2 Comments
This manuscript presents results from a retrospective study providing evidence that the use of chest CT images to determine muscle mass may be beneficial for predicting early postoperative outcomes in pediatric patients undergoing Fontan procedures. The results have potential clinical significance and the article is likely to be of interest to your readership. I have the following major comments regarding the manuscript:
Point 1. Sarcopenia by definition is the age-related loss of skeletal muscle mass and function. It is a geriatric condition and therefore not observed in children. Furthermore, the investigators have only examined muscle mass, and not function. Therefore the title, abstract and wording throughout the paper should refer only to low muscle mass and not sarcopenia.
Response 1: Thank you for this constructive comment. We agree with the reviewer’s concern and have acknowledged the issue regarding the definition of sarcopenia. Recently, sarcopenia has been reported in pediatric patients with chronic or critical illness [JPEN J Parenter Enteral Nutr. 2019 Jul 22. doi: 10.1002]. It has been recognized that sarcopenia may occur earlier in life and the sarcopenia phenotype can be attributed to various factors besides age [Age Ageing. 2019 Jan 1;48(1):16-31]. However, as the reviewer pointed out, we were unable to meet all the criteria for sarcopenia owing to the nature of our retrospective study design. Therefore, we revised the term “sarcopenia” to “muscle wasting” throughout the manuscript considering that we only evaluated muscle mass.
Point 2. I do not see the value of including a control group of children who underwent the Nuss procedure. These patients do not seem to be comparable to the Fontan group. The Nuss group is a mean of 5 years older than Fontan (mean age 8 vs 3 years), and the upper age range for Fontan falls below the lower age range for Nuss. Although muscle mass estimates are normalised to body surface area, there is still an assumption that proportions of muscle mass are similar in younger children and older children, which may not be the case. Age-adjusted results are provided but this potentially doesn't control for all the factors that may explain better muscle mass in older children.
Point 3. Further to the above two points, I do not believe it is appropriate to define low muscle mass cut-offs using the Nuss group as the reference population. In addition to the fact that they do not appear to be comparable populations, existing low muscle mass cut-offs for older adults are generally based on values around ~<2SD below the mean for healthy young adult populations. The authors of this manuscript have used the median for older Nuss patients to identify low muscle mass in younger Fontan patients, and this is not based on evidence. I therefore believe the analyses should focus on continuous values for muscle mass rather than arbitrary dichotomous variables
Point 4. In summary, I do not believe that the data for the Nuss group adds value to the paper. It therefore could be removed so that the focus is on the Fontan group only. If cut-offs were to be used for low muscle mass, I think it would be of more interest to determine cut-offs associated with poorer prognosis from within the Fontan population; this data would be of greater utility to clinicians.
Response 2, 3, 4: Thank you for the valuable comment. We agree with the reviewer’s concern regarding the control group. Therefore, we excluded the Nuss group and focused on the Fontan group in the revised manuscript. Patients undergoing the Fontan operation were divided into the quintiles of total skeletal muscle index (TSMI). Muscle wasting group was defined as a TSMI value lower than the median TSMI for the second quintile. Methods, Result, and Conclusions were revised following the reviewer’s comment.
Other comments:
Point 5. The odds ratios in the abstract and table need to include units for the independent variables.
Response 5: We revised the abstract and tables following your comment. Abstract now reads like;
“Multivariable analyses revealed that TSMI was associated with an increased duration of intensive care unit (>5 days) and hospital stay (>14 days) (odds ratio [OR] 0.86; 95% confidence interval [CI] 0.77–0.96; P = 0.006 and OR 0.92; 95% CI 0.85–0.99; P = 0.035) and incidence of major postoperative complications (OR 0.90; 95% CI 0.82–0.99; P = 0.039).”
Point 6. The abstract should state that this is a retrospective study.
Response 6: In order to fulfill the reviewer’s comment, we added this to the abstract as follows.
“The electronic medical records of 74 patients (aged <18 years) who underwent the Fontan operation were retrospectively reviewed.”
Point 7. Is data available on the intra- and inter-operator reliability of the CT measurements?
Response 7: This is an excellent comment which was appreciated by the authors. As recommended, we added this content in the methods and results as follows.
In the Methods section:
“Intraobserver and interobserver variability were estimated using the intraclass correlation coefficient.”
In the Results section:
“The intraclass correlation coefficient for CT measurements was 0.986 (P < 0.001), which was in the acceptable range.”
Point 8. Was a power calculation performed to determine whether the sample size is adequate?
Response 8: We thank you for this constructive comment. We performed the power calculation for the current study post-hoc, given its retrospective design. The primary endpoint of this study was to identify the effect of muscle wasting on the incidence of major postoperative outcome. Considering the incidence of major postoperative complication in total study subjects was 28% and that in the muscle wasting group was 48%, the results for post-hoc power calculation were as follows, according to the power calculator available online (https://clincalc.com/stats/Power.aspx).
Point 9. There appears to be an error in the description of the study population beginning on line 125; it states that "194 pediatric patients who underwent the Fontan operation between January 2006 and December 2016, 135 were included in the study". But the 194 appears to be for both Fontan and Nuss, as only 74 Fontan were included.
Response 9: We revised the results and Figure 2 following the reviewer’s comment.
In the Results section:
“Of the 194 pediatric patients who underwent the Fontan operation between January 2006 and December 2016, 74 were included in the final analysis (Figure 2).”

Round 2
Reviewer 1 Report
Thank you for addressing my comments. The work up for a Fontan is clearly different in your experience.
Please add this portion of your response from question two to the actual manuscript.
"Our institution conducts a CT scan as a routine assessment before the Fontan operation. However, if necessary, cardiac catheterization is also performed to obtain information."
Author Response
Response to Reviewer 1 Comments
Thank you for addressing my comments. The work up for a Fontan is clearly different in your experience.
Point 1: Please add this portion of your response from question two to the actual manuscript.
"Our institution conducts a CT scan as a routine assessment before the Fontan operation. However, if necessary, cardiac catheterization is also performed to obtain information."
Response 1: Thank you for the valuable and positive comments. We added this content in the materials and methods section following your comment.
“Our institution conducts a CT scan as a routine assessment before the Fontan operation. However, if necessary, cardiac catheterization is also performed to obtain information."

Reviewer 2 Report
I thank the authors for their considered responses to my previous comments. I have the following additional recommendations:
1. I think it is important to use the term "low muscle mass" rather than "muscle wasting" throughout the article. Given the very young age of the study population, they have not yet achieved peak muscle mass and it is unlikely that these children are undergoing muscle wasting per se; rather, they likely have low rates of muscle accrual since birth. As such I believe you should only use the term low muscle mass (as you have in the article title).
2. The data from the post-hoc power calculation does not appear to be provided either in the author response or in the manuscript. I believe the data from this power calculation should be provided in the limitations section of the discussion.
3. The sentence in the abstract, "Multivariable analyses revealed that TSMI was associated with an increased duration..." should be revised as follows: "Multivariable analyses revealed that higher TSMI was associated with lower likelihood of an increased duration...".
4. The previous request to include units of the independent variable in the abstract has not been appropriately addressed. The format should be as follows: "odds ratio [OR] 0.86; 95% confidence interval [CI] 0.77–0.96; P = 0.006 and OR 0.92; 95% CI 0.85–0.99; P = 0.035 per 1 cm2/m2 increase in TSMI).
Author Response
Response to Reviewer 2 Comments
I thank the authors for their considered responses to my previous comments. I have the following additional recommendations:
Point 1: I think it is important to use the term "low muscle mass" rather than "muscle wasting" throughout the article. Given the very young age of the study population, they have not yet achieved peak muscle mass and it is unlikely that these children are undergoing muscle wasting per se; rather, they likely have low rates of muscle accrual since birth. As such I believe you should only use the term low muscle mass (as you have in the article title).
Response 1: We thank you for this comment which was well taken by the authors. We agree with the reviewer’s concern. We revised the term “muscle wasting” to “low muscle mass” throughout the manuscript.
Point 2: The data from the post-hoc power calculation does not appear to be provided either in the author response or in the manuscript. I believe the data from this power calculation should be provided in the limitations section of the discussion.
Response 2: Thank you for the valuable comment. We understand the potential limitation of the current study on the issue raised by the reviewer and added this shortcoming in the limitation section more clearly. Now it reads in the limitation section as follows;
“Finally, we performed a post-hoc power analysis for the sample size, given the retrospective design of this study. The incidence of major postoperative complications in total study patients was 28% and that in the low muscle mass group was 48%. The sample size of 74 resulted in post-hoc power of 95.4%, considering an α-error of 0.05.”
Point 3: The sentence in the abstract, "Multivariable analyses revealed that TSMI was associated with an increased duration..." should be revised as follows: "Multivariable analyses revealed that higher TSMI was associated with lower likelihood of an increased duration..."
Response 3: This is an excellent comment, which was appreciated by the authors. As recommended, we changed the expression as follows;
“Multivariable analyses revealed that higher TSMI was associated with lower likelihood of an increased duration……”
Point 4: The previous request to include units of the independent variable in the abstract has not been appropriately addressed. The format should be as follows: "odds ratio [OR] 0.86; 95% confidence interval [CI] 0.77–0.96; P = 0.006 and OR 0.92; 95% CI 0.85–0.99; P = 0.035 per 1 cm2/m2 increase in TSMI).
Response 4: Thank you for this thoughtful comment. We revised the abstract following your comment. Abstract now reads like;
“odds ratio [OR] 0.86; 95% confidence interval [CI] 0.77–0.96; P = 0.006 and OR 0.92; 95% CI 0.85–0.99; P = 0.035 per 1cm2/m2 increase in TSMI) and incidence of major postoperative complications (OR 0.90; 95% CI 0.82–0.99; P = 0.039 per 1cm2/m2 increase in TSMI).”
